# Temperature Fluctuations Compensation with Multi-Frequency Synchronous Manipulation for a NV Magnetometer in Fiber-Optic Scheme

**DOI:** 10.3390/s22145218

**Published:** 2022-07-12

**Authors:** Ning Zhang, Qiang Guo, Wen Ye, Rui Feng, Heng Yuan

**Affiliations:** 1Research Center for Quantum Sensing, Intelligent Perception Research Institute, Zhejiang Lab, Hangzhou 310000, China; guoqiang@zhejianglab.com; 2Division of Mechanics and Acoustic Metrology, National Institute of Metrology, Beijing 100029, China; yewen@nim.ac.cn; 3The School of Instrumentation and Optoelectronic Engineering, Beihang University, Beijing 100191, China; ruifeng@buaa.edu.cn; 4Hangzhou Innovation Institute, Beihang University, Hangzhou 310000, China

**Keywords:** temperature fluctuations compensation, multi-frequency synchronous manipulation, NV center, magnetic field sensing, fiber sensor

## Abstract

Nitrogen-vacancy (NV) centers in diamonds play a large role in advanced quantum sensing with solid-state spins for potential miniaturized and portable application scenarios. With the temperature sensitivity of NV centers, the temperature fluctuations caused by the unknown environment and the system itself will mix with the magnetic field measurement. In this research, the temperature-sensitive characteristics of different diamonds, alongside the temperature noise generated by a measurement system, were tested and analyzed with a homemade NV magnetometer in a fiber-optic scheme. In this work, a multi-frequency synchronous manipulation method for resonating with the NV centers in all axial directions was proposed to compensate for the temperature fluctuations in a fibered NV magnetic field sensing scheme. The symmetrical features of the resonance lines of the NV centers, the common-mode fluctuations including temperature fluctuations, underwent effective compensation and elimination. The fluorescence change was reduced to 1.0% by multi-frequency synchronous manipulation from 5.5% of the single-frequency manipulation within a ±2 °C temperature range. Additionally, the multi-frequency synchronous manipulation improved the fluorescence contrast and the magnetic field measurement SNR through an omnidirectional manipulation scheme. It was very important to compensate for the temperature fluctuations, caused by both internal and external factors, to make use of the NV magnetometer in fiber-optic schemes’ practicality. This work will promote the rapid development and widespread applications of quantum sensing based on various systems and principles.

## 1. Introduction

Sensing and metrology play important roles in various fields, and recent advanced development in quantum sensing technology has enabled new possibilities for related applications [1]. Magnetic field measurements using the nitrogen-vacancy (NV) center in diamonds have advantages in high sensitivity, stable chemical properties, short response times, and low toxicity [2,3]. Based on the optical excitation and detection of the NV center, the techniques have significant potential in terms of volume and sensitivity when forming a fiber magnetic field sensor [4,5]. At present, nanodiamond and bulk diamond have achieved intracellular measurement and electromagnetic imaging combined with fiber, and an NV magnetometer in fiber-optic scheme has been developed with high resolution and high stability [6,7,8].

Owing to the characteristics of the crystal lattice structure of diamond, the NV center has a strong temperature sensitivity [9,10]. Therefore, during the magnetic field measurement, the temperature change of the measured environment and the internal temperature noise of the sensor system will cause resonance frequency shift and spin state instability. The temperature fluctuations will seriously affect the sensitivity and the stability of the magnetic field measurement [11,12,13]. In the classic confocal scheme, S. K. R. Singam and A. M. Wojciechowski adopted the frequency jump and driving frequency modulation method for magnetic field measurement, which effectively suppressed the influence of temperature fluctuations [14,15]. Y. Hatano introduced a centimeter-scale portable quantum sensor head-in-fiber scheme. A continuous measuring mode is adopted to analyze the temperature noise from the magnetic field spectrum and to test long-term stability under a magnetic shielding condition in a laboratory environment [6]. For miniaturized, portable fiber magnetometers, the compact application scenarios make it difficult to diffuse the heat generated by the measurement system or the biochemical process in living organisms, and adaptive in situ compensation methods for temperature drift are particularly needed [16].

In this work, an NV magnetometer-in-fiber-optic scheme was established, with diamond particles and homemade microwave coils suppressing the sensor head to several millimeters. The temperature sensitivity of NV centers in deferent diamond samples was analyzed. We also directly measured the temperature noise caused by lasers and microwave systems in the magnetic sensing system. During the magnetic field measurement process, the multi-frequency synchronous manipulation method was used to monitor the all-states spin resonance and compensate for the temperature drift, which effectively decreased the influence of temperature changes and increased the fluorescence contrast and measurement stability. A magnetic field measurement that is robust to temperature fluctuations without external equipment will be beneficial to the micro-magnetic field measurement scene and to biological sensing. The work will promote the improvement of measurement and metrology using quantum sensing technology.

## 2. Methods

NV centers are crystalline imperfections in diamond lattices. An NV center consists of a nitrogen atom substituting for a carbon lattice site that is bound to an adjacent vacant lattice site, as shown in Figure 1a. The ground state and the excited state of an NV center at room temperature are both spin triplets with symmetric energy levels of ^3^A_2_ and ^3^E, there being metastable states ^1^A^1^ and ^1^E between the two, as shown in Figure 1b [2,17]. The spin-spin interaction of the two unpaired electrons in the ground state causes zero field splitting (ZFS) between the m_s_ = 0 and m_s_ = 1 states, and the splitting energy level D is approximately 2.87 GHz. The process of the ^3^E excited state returning to the ^3^A^2^ ground state is accompanied by the release of red-band fluorescence with a zero-phonon line at 637 nm. At the same time, the phonon sideband also emits fluorescence within the range of 637–800 nm. Based on the specific energy level transition rules of the NV center, the excitation laser polarizes the NV center electron spin into the m_s_ = 0 state [18,19]. With the microwave magnetic field at an appropriate frequency, resonating with the electron spin, the fluorescence intensity emitted by different spin states will be different. By detecting the fluorescence change in the red-band wavelength region, the magnetic field can be sensed and measured [20].

A high-concentration HTHP diamond particle with a diameter of approximately 300 μm was used in the temperature fluctuations experiments, which were irradiated at 1 × 10^18^ e^−^/cm^−2^ and annealed at 800 °C for 2 h. The diamond particle was adhered to the fiber tip by UV glue, and the excitation laser transmission and the fluorescence collection were achieved with the same multimode fiber. The 532 nm excitation laser achieved pulsed modulation with an acousto-optic modulation (AOM, G&H 3200-126). The collected fluorescence returned to the dichroic mirror through the multimode fiber and was detected by a detector (Thorlabs, APD120A/M) with a long-pass filter. The microwave signal (Minicircuits, SSG-6000RC) was connected to a designed microwave conversion board after passing through the microwave switch and the power amplifier. The conversion board transmitted the pulse-controlled microwave signal to a gold wire with a diameter of 100 µm. The gold wire was tightly wound into a spiral on the optical fiber tip, with the diamond particle at the spiral center. The head and tail of the gold wire were connected to the joints of the conversion board. The head was connected to the microwave input signal through the conversion board, and the tail was connected to a 50 Ω load. Homemade three-dimensional Helmholtz coils provided the specific magnetic field required for the experiments. The temperature change was achieved via the use of customized heating films and a temperature controller. The gold wire was placed under the fiber head and was not in contact with the fiber head during the experiment. This was carried out to avoid the displacement of the measuring head caused by the thermal expansion and contraction. The experimental setups are shown in Figure 1c.

The temperature dependence of both the bulk and particle diamonds were studied with optical detection magnetic resonance (ODMR) experiments at variable temperatures with a liquid nitrogen thermostat (Janis ST-300MS). The thermostat provided a constant temperature condition in the range of 80–300 K and a temperature stability better than 50 mK. The light-passing window and the microwave feed device in the liquid nitrogen thermostat were customized and improved according to the requirements of the NV center confocal optical path and the microwave control experiments. The single crystal bulk CVD diamond from Element Six with 800 ppb original N concentration was irradiated with electrons at 5 × 10^17^ e^−^/cm^−2^ and annealed at 800 °C for 2 h.

## 3. Results

### 3.1. Temperature Dependence of NV Centers in Different Diamonds

The zero-field splitting (ZFS) is closely related to the lattice structure of a diamond and the distance between two unpaired electrons. Owing to the different growth environment and preparation processes, there will be fine differences in the lattice structure, even for different regions in the same sample [21]. Therefore, the sensitivity of each diamond in this research to temperature differed. For sensing and metrology applications, the temperature-dependent calibration of the diamond samples used in the experiments was essential. To calibrate the ZFS energy D of different diamond samples, the ODMR experiment with a zero-bias magnetic field was performed by changing the thermostat working temperature. The resonance frequencies corresponding to the NV centers at different temperatures were then extracted. Figure 2a shows the variation trend of the ZFS energy D of the NV center in the HTHP diamond particles with temperature. With the increase in the set temperature of the liquid nitrogen thermostat, the resonance peak of the NV centers continued to shift to the left; that is, the ZFS energy D was negatively related to the temperature.

The temperature change experiment was performed on both the HTHP particles and the bulk CVD samples to obtain the corresponding temperature-sensitive rules. The resonance frequency of the NV centers for zero magnetic field was extracted, as shown in Figure 2b. The experimental results for the HTHP samples at the same temperature had larger errors than those for CVD samples, but the overall trend of the ZFS energy D with temperature was essentially the same. Referring to the previous research, the following formula was adopted to fit the experimental results of the ZFS energy D with temperature, as shown in the solid lines in Figure 2b.
(1)$$D\left(T\right)=D_{0}-\frac{\alpha T^{4}}{{\left(T+\beta \right)}^{2}},$$
where $D_{0}$ is the ZFS energy corresponding to absolute zero, and α and β are the coefficients related to the particular sample [11,21,22]. The HTHP particles varied in local lattice structure. The growing condition of the CVD diamond samples was similar, resulting in a smaller difference in lattice structure than the HTHP diamond samples and fewer resonance frequency errors. The fitting error in formula (1) of CVD diamond samples was considered to be the result of numerical errors coming from theoretical formula simplification. To facilitate the applications of the magnetometer in the room temperature range, the results of the fitting curve can be further divided into two sections. In the low temperature range below 240 K, the zero-field splitting energy change law is nonlinear. In the near room temperature range of 240–300 K, the change law is basically linear. In the linear region, the ZFS energy D change rate of the HTHP samples with temperature is dD/dT=−70.2 kHz/K, while the change rate of the CVD samples is dD/dT=−67.9 kHz/K.

### 3.2. Thermal Noise Generated by the Measurement System

In the process of magnetic field measurement using an NV center electron spin ensemble, temperature fluctuations caused by the experimental environment and the measurement system will mingle with the detection signal and affect the measurement sensitivity and stability [23,24]. In an NV center magnetic field measurement system, the main sources that may cause temperature fluctuations around the diamond include the thermal effects of the longtime laser irradiation and the high-frequency microwave magnetic field. To assuage this concern, the ODMR experiments with microwave power scanning were performed on the fiber measurement scheme of the HTHP diamond particles. The 532 nm laser power in front of the dichroic mirror was increased from 10 mW to 80 mW. The experimental results showed that with the increase in the laser power, the width of the resonance peak of the NV center increased, but the resonance frequency experienced no significant shift. This showed that for the high-concentration HTHP samples and the experimental fiber systems used in this research, the thermal effect caused by the laser power was not obvious. This might have been because the laser loss of the fiber was large, or because the thermal effect caused by the laser power was too weak to observe.

Under common conditions, the temperature near the surface of a microwave resonator should be positively correlated with the input microwave power. For different microwave power, the Rabi oscillation frequency formed by the NV center was different. The π pulse of the different microwave power was determined first, and the corresponding ODMR was then obtained. As shown in Figure 3a, there was an increase in the input microwave power, before the amplifier shifted the resonance frequency of the NV center to a lower frequency. This was consistent with the expected result—that the increase in microwave power would produce a thermal effect. The relationship between the resonance frequency of the NV center and the microwave power is shown in Figure 3b. The nonlinear relationship between the microwave input power and the resonance frequency was fitted as $\frac{A}{P^{2}}+e^{BP}+c$, with P representing the microwave input power. The relationship was supposed to change slightly depending on the microwave resonator with the negative correlation.

### 3.3. Multi-Frequency Synchronous Manipulation Scheme

The influences of the magnetic field and the temperature on the ODMR spectrum of the NV center were different. The method for suppressing fluctuations in temperature could be established according to the difference influence rules [15,25]. The fiber magnetometer here used [100] HPHT diamond particles. Since the NV center could provide vector magnetic field measurement, a bias magnetic field in the [111] direction was applied to extract the NV centers with the same crystal orientation from the ODMR spectrum. As shown in Figure 4, the resonance lines were divided into two symmetry groups. The deeper peaks in the figure represent the NV centers in the non-[111] direction and the shallower peaks represent the NV centers in the [111] direction.

In the magnetic field measurement mode, the temperature change caused the ODMR line to shift in the same direction. For instance, when the temperature decreased, the ODMR shifted left, which was shown as the red resonance spectrum in Figure 4a. The fluorescence change trend at the points symmetrical with the ZFS energy D (represented by $f_{a}$ and $f_{b}$) was theoretically the opposite. Therefore, by simultaneously manipulating the specific frequency with opposite changing trends, the external factors that caused the fluorescence fluctuations would be canceled, and then the temperature effect would not be introduced in the final magnetic field measurement signal. For the magnetic field measurement scheme, the two slopes of $f_{a}$ and $f_{b}$ could be simultaneously dual frequency manipulated, and the fluorescence change could be expressed by a formula written as
(2)$$dF=\left({\left.\frac{\partial F}{\partial B}\right|}_{f_{a}}+{\left.\frac{\partial F}{\partial B}\right|}_{f_{b}}\right)dB+\left({\left.\frac{\partial F}{\partial T}\right|}_{f_{a}}+{\left.\frac{\partial F}{\partial T}\right|}_{f_{b}}\right)dT,$$
where each component represents the change rate of the fluorescence spectrum with the magnetic field *B* or the temperature *T* at the $f_{a}$ and $f_{b}$ frequencies. As a result, the ODMR shift and the resonance frequency fluctuations caused by temperature can be eliminated, and the signal-to-noise ratio of the magnetic field measurement can be improved, to achieve the suppression of the temperature effect. In order to take full advantage of the NV centers in four lattice directions, the simultaneous dual-frequency manipulation was performed at the resonance lines of the non-[111] direction and the [111] direction. The magnetic field sensing scheme used here was the microwave frequency modulated method. As shown in Figure 4b, the microwave field of resonance lines in the non-[111] direction was sinusoidally modulated as *f*_MW1_(t), and the detected fluorescence *F*_1_(t) was thereby sinusoidally modulated correspondingly. By simultaneously detecting the modulated fluorescence signal of both the non-[111] direction and the [111] direction, the equivalent magnetic field change caused by the microwave field could be resolved.

Since the diamond sample and the fiber tip were both non-metallic, the thermal expansion and contraction effect produced by the temperature change was small, and the influence of these effects on the stability of the optical system could be ignored. At room temperature, only the set temperature of the heating film was changed, looping between 29 °C and 31 °C, while the microwave frequency modulation center was kept at the maximum slope of the selected resonance lines to record the fluorescence change. The magnetic field was measured using the amplitude modulation of the microwave frequency [15,26]. The modulation frequency of the four detected microwave frequencies was 0.1 Hz, and the modulation amplitude was 1 MHz. The experiment was pulse-controlled, and each pulse cycle modulated the microwave signal once. The fluorescence signal that was affected by temperature changes is shown in Figure 5. The temperature first rose and then fell: the red line indicates that the fluorescence at $f_{a}$ gradually rose until the temperature stabilized, after which it then gradually decreased. The green line indicates that the fluorescence at $f_{b}$ changed in the opposite manner. As a result, after verifying the influence of the temperature changes on the shift of the ODMR spectrum, the dual-frequency manipulation at $f_{a}$ and $f_{b}$ was applied to the NV center ensemble. The heating film temperature setting and fluorescence signal were the same as before. The blue line in Figure 5 represents the detected fluorescence of the dual-frequency manipulation at $f_{a}$ and $f_{b}$. In the simultaneous measurements at the symmetry frequency points, the fluorescence change was found to be significantly reduced during the heating and cooling process. The dual-frequency manipulation was also performed at $f_{c}$ and $f_{d}$, and the temperature compensation effect was the same. The detected fluorescence signal amplitude was relatively slight because there were only 1/4 NV centers along the [111] axis in the natural diamond.

To further increase the number of NV centers involved in the measurement, the simultaneous manipulation method based on the ODMR spectrum was extended to multiple NV axles [27,28]. The multi-frequency manipulation was performed simultaneously at the selected four frequencies $f_{a}$ to $f_{d}$ to produce the temperature compensation effect for the NV centers in both the [111] and non-[111] axles. The black line in Figure 5 indicates that the multi-frequency simultaneous manipulation had obvious effectiveness, with the increased fluorescence signal and measurement stability. According to the modulation parameters, the detected fluorescence signal was fitted, and the fluorescence change rate was defined
(3)$$\kappa =\frac{std\left(F-\tilde{F}\right)}{mean\left(F\right)}$$
where F is the detected fluorescence signal, and $\tilde{F}$ is the theoretical fluorescence signal fitted according to the modulation signal. The results showed that the fluorescence change rate κ of the multi-frequency manipulation was reduced to 1.0% from 5.5% of the single-frequency manipulation within the temperature range of ±2 °C. By simultaneously manipulating the NV centers at the symmetry frequency points of the resonance lines, the fluorescence signal changes caused by the temperature fluctuations were effectively suppressed. The multi-frequency synchronous manipulation method maximized the fluorescence contrast through the omnidirectional manipulation scheme, which improved the SNR of the magnetic field measurement signal of 44.7% compared to the single-frequency manipulation at $f_{a}$.

## 4. Discussion

The temperature dependence of CVD bulk diamond and HTHP diamond particles was investigated in this research, and the experimental results had a small amount of deviation. The CVD diamond samples used in this research had low and similar nitrogen concentrations, the same sample preparation process, and relatively uniform lattice structures and atomic distances. Therefore, the measurement result differences of the ZFS were not obvious. The nitrogen concentration distribution of the HTHP diamond particles had a relatively large gap, which resulted in bigger experimental errors. The NV centers in diamond are the system core of the quantum fiber magnetometer. Thus, the accurate generation of NV centers and the accurate calibration of the main parameters including temperature dependence are topics worthy of in-depth study.

The multi-frequency synchronous manipulation scheme was verified to be feasible and effective, which left 1.0% fluorescence noise in the modulated magnetic field measurement. The remaining 1.0% fluorescence noise was equivalent to the fluorescence signal change rate obtained with modulated measurement when the temperature was constant. This showed that the fluorescence signal fluctuations caused by other factors such as the laser power fluctuations, microwave power fluctuations, and detector noise in the system were around 1.0%. Therefore, to further improve the sensitivity and the stability of the NV magnetometer, it will be necessary to focus on the stability analysis and robust control of device components in future research, which is also the essential way to determine the applications of miniaturized NV center sensors.

## 5. Conclusions

The temperature sensitivity investigation and the analysis of different diamond samples were conducted using a liquid nitrogen constant temperature microscopy experiment system. Based on the different sensitivity characteristics of the NV centers to the magnetic field and the temperature, a magnetic field measurement scheme that compensated for the temperature noise was proposed, and the signal intensity of fluorescence detection and the SNR were improved based on the multi-frequency synchronous manipulation. In the experiment, the fluorescence signal fluctuations caused by the temperature change were reduced from 5.5% to less than 1.0%, which effectively reduced the influence of the temperature fluctuations in the environment and the measurement system. This work will help improve the stability and practicability of NV magnetometer in fiber-optic schemes, making them more suitable for compact application scenarios and living organisms. It will also provide a reference for the practicality and development of related quantum sensing technology [29].

## Figures and Tables

**Figure 1 sensors-22-05218-f001:**
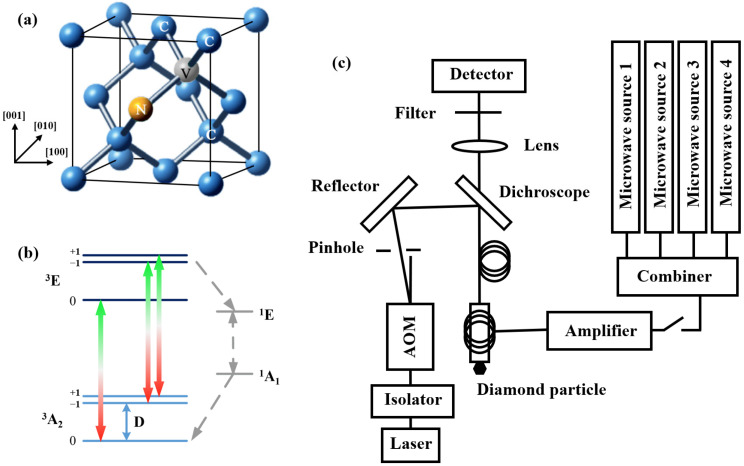
(**a**) The spatial structure of an NV center in a diamond crystal lattice. (**b**) The energy level structure of the NV center with S = 1. (**c**) Block diagram of experimental system of the fiber magnetometer with NV centers.

**Figure 2 sensors-22-05218-f002:**
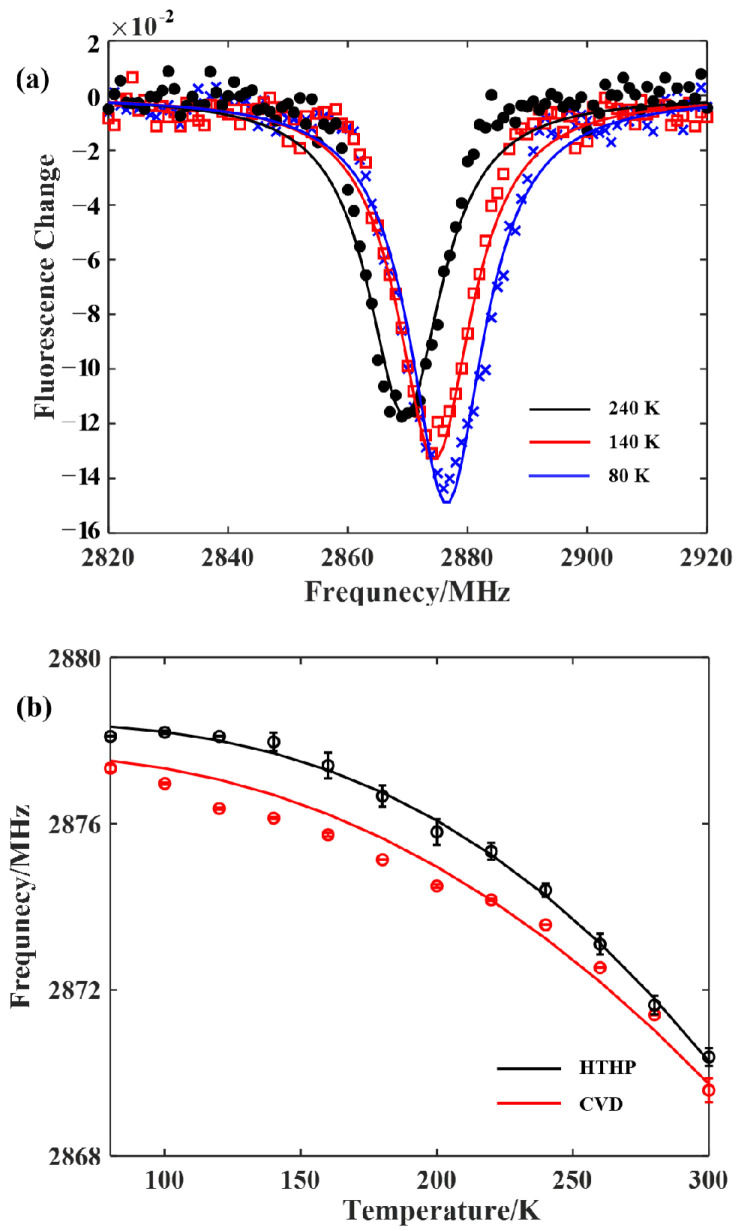
(**a**) The shifts of the ODMR resonance lines of the NV centers in the HTHP diamond particles with the temperature. (**b**). The resonance frequency changes of the NV centers with temperature in both the HTHP diamond particles and the CVD bulk diamond.

**Figure 3 sensors-22-05218-f003:**
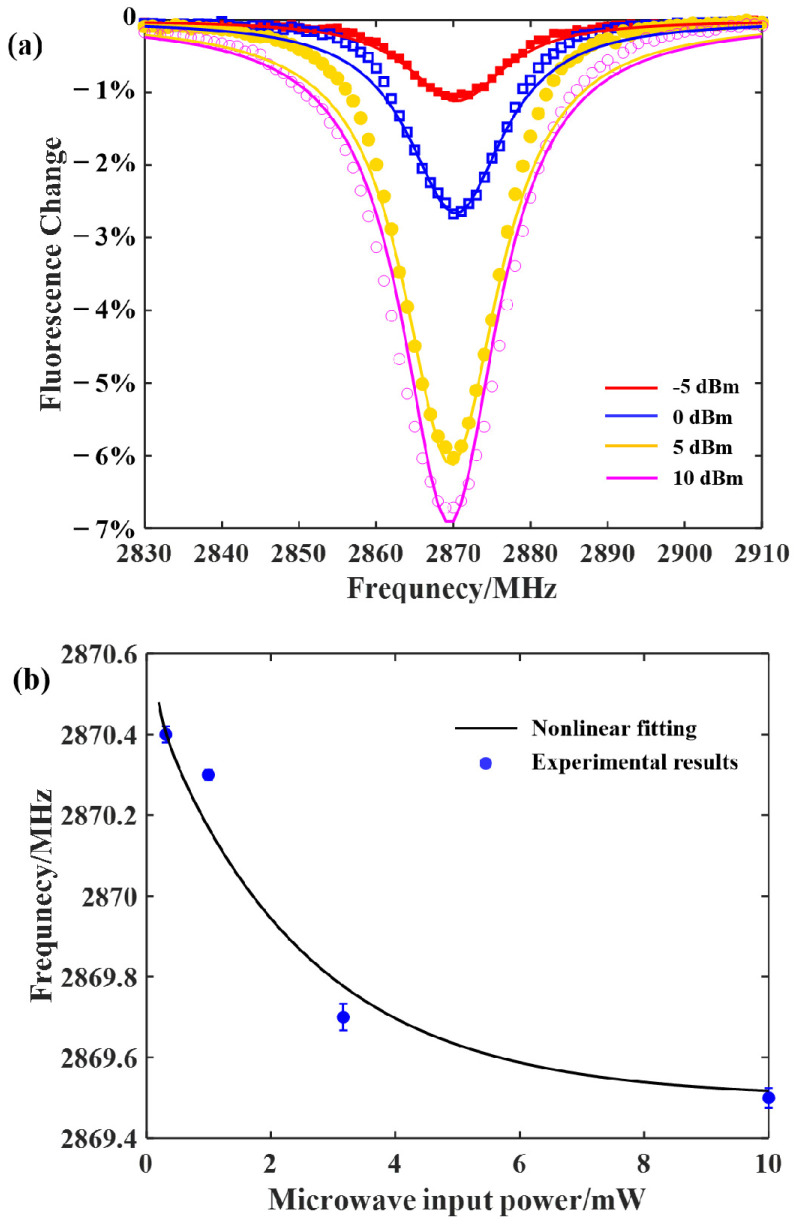
(**a**) The ODMR resonance lines shifted with the different microwave input power. (**b**) The changes of the resonance frequency of the NV centers with the microwave input power. The red line and the black line are the linear fitting and the nonlinear fitting of the results.

**Figure 4 sensors-22-05218-f004:**
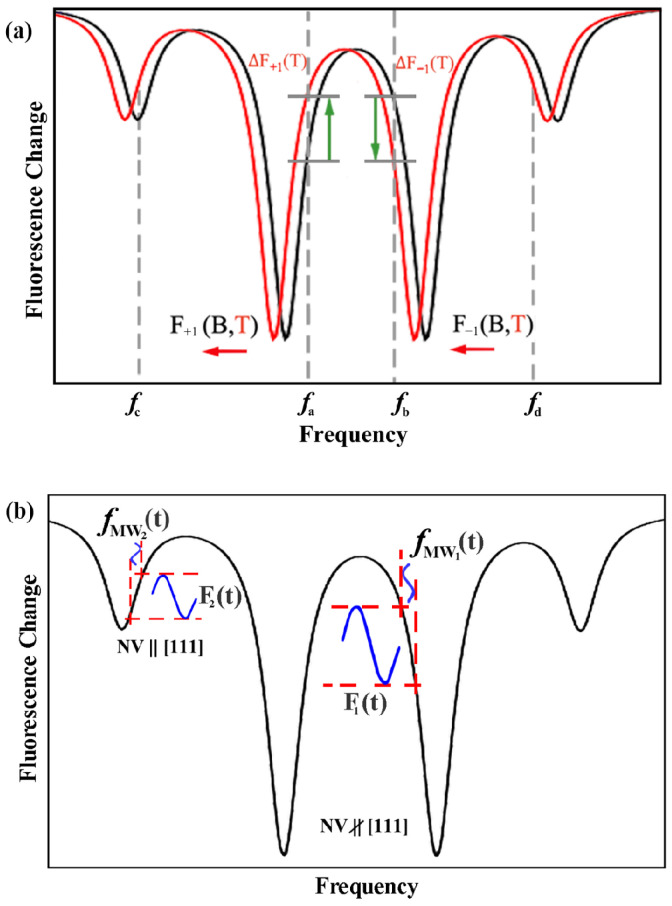
(**a**) The schematic diagram of the dual-frequency simultaneous manipulation depending on different reactions of the temperature and the magnetic field variations of the ODMR resonance lines of the NV centers. (**b**) The dual-frequency simultaneous manipulation for the NV center in all axial directions.

**Figure 5 sensors-22-05218-f005:**
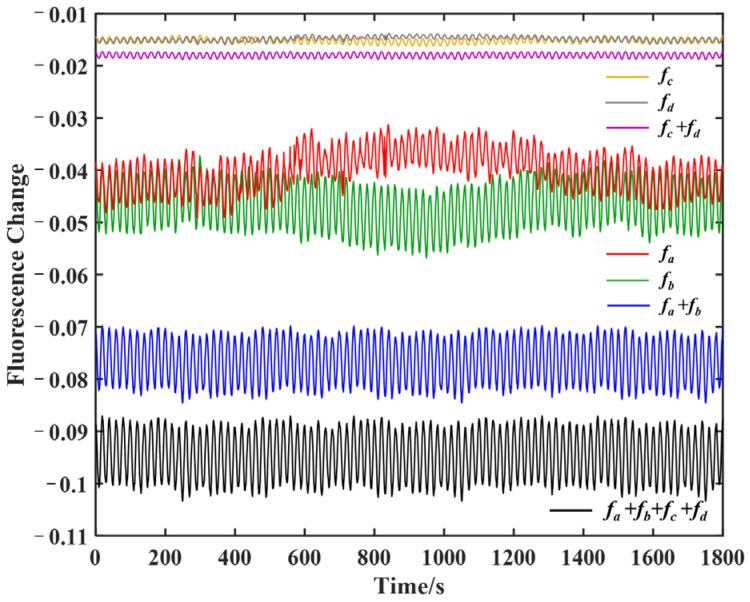
The detected fluorescence signals of the modulated magnetic field sensing experiments with single-frequency, dual-frequency, and multi-frequency manipulation scheme. The anomaly signal fluctuations were caused by the set temperature change.

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
