# Peer review of "Temperature Fluctuations Compensation with Multi-Frequency Synchronous Manipulation for a NV Magnetometer in Fiber-Optic Scheme"

_sensors, 2022, doi:10.3390/s22145218_

Round 1

Reviewer 1 Report

The manuscript provides a description of NV- thermometry conducted in a micro-scale diamond attached to the end of a optical fibre.   The authors report on the laser dependence, microwave dependence on the position of NV- ground state zero field splitting (D), and attribute the observed changes to temperature effects.  The authors propose a method to account for temperature variations in D from monitoring multiple frequencies of an ODMR experiment and demonstrate this to show a reduction in temperature induced inaccuracies between an applied and measured magnetic field.

The results presented in the manuscript don’t work together to produce a clear nor impactful conclusion.  The demonstrated temperature-based noise reduction is not suitable to ascertain its use for optical fibre-based NV- magnetometry.  The reported observations of the temperature dependence of the NV- zero field splitting measured by ODMR is well covered in the literature, and no new insights are provided in this manuscript.  The analysis methods employed are overly simplified.  The reason behind the inclusion of a CVD diamond is unclear.  The purpose of extracting linearised parameters for dD/dT and dF/dPmw is unclear.

Specific comments:

Line 36. It is unnecessary to cite an arvix paper to establish the widely reported prospects for NV- magnetometry.  Peer review work would be more appropriate.

Fig1:  The lineshapes of fits to ODMR ZFS are undefined, and that the solid lines are fits is unlabelled.  The source of errors displayed is not described.  The fit formula (1) to the CVD diamond data disagrees well outside the displayed error, could the authors provide an explanation discrepancy?

Ln 148:  Formula 1 indicates that  D(T) is expected to be parabolic at higher temperatures (confirmed in ref 22), what is the purpose of describing it as a linear function?  What is the expected error on this approximation and would that be consequential for the intended purpose?

Fig2: Fits not described.  Not clear that there isn’t a slight shift in ZFS ODMR peak to higher power.  The plot (b) for microwave provides a better visitation of any peak position changes.  Why is there no error displayed on experimental results in plot b?  Especially given the small size of the shifts displayed, is the fit trend significant?  What is the purpose of describing it as a linear function.  What nonlinear function is fit to provide the black line.

Can the authors comment on the magnitude of the estimated heating from driving the microwave resonator?  Is it reasonable compared to the power input and cooling power of the thermostat?

Fig5:  Could the [111] aligned NV B-field detection be plotted on a more appropriate scale, perhaps in a second plot?

The name “fluorescence change rate ?” is potentially confusing, as it is a measure of the detector error in reporting the applied signal.  It not comparable to alternative modes of minimising errors and thus does not provide a good figure of merit for assessing the improvement provided.

By adding the measured frequencies fa and fb the authors are correct to identify that temperature variations are cancelled.  However, in this mode the effect of any sources of fluorescence variation that are not due to magnetic field changes is doubled.  Presumably in the presented experiment sources of background fluorescent signal is well controlled, however given this trade off it is unclear if  this measurement regime would provide a material benefit in more realistic environments.

With four microwave lines, i.e. four frequencies to be monitored a much more robust scheme can easily be envisioned.  Such as if instead of monitoring the [111] direction, those two frequencies were used to monitor the other side of the ODMR peak.  Then (fa+fb) – (fc+fd) would act to both account for common shifts due to thermal effects, but common mode rejection of non-ODMR based variations in fluorescence.

What is the noise floor of the detector as presented.  How does it compare to previous demonstrations?

Reviewer 2 Report

The manuscript by Ning Zhang et al. deals with an important field of NV diamond sensors. The authors describe the method of mitigating temperature instabilities by special addressing of different components of ODMR signals in two samples of NV diamond attached to an optical fiber and placed in a magnetic field.

I cannot recommend the publication of this manuscript because of the following reasons:

  • The title containing “a fiber NV magnetometer” is misleading. As a “fiber NV magnetometer”, most people understand the sensor where micro- or nano-diamonds with NV centers are inserted into the fiber material such that the sensing takes place within the fiber. Recent examples of such sensors are APL Mater.8, 081102 (2020) and Carbon 196, 10 (2020). The approach described in the manuscript is just a “fiberized” sensor, where an NV particle or a bulk NV diamond is simply linked by an optical fiber to other parts of the device (relevant examples are: Sci. Rep. 4, 5362 (2014), Frontiers in Photonics 2, 732748 (2021) ). Consequently, it is not very original, the described reduction of the temperature and/or magnetic field instabilities does not really differ from what has been described with bulk samples, e.g. in Ref.15.  By the way, Ref.15 is incorrectly cited: the correct reference is Appl.Phys.Lett. 113, 013502 (2018).

  • Starting from the title and throughout the text, the authors write about “synchronous frequency manipulation” without specifying what is meant by the “manipulation” and what is “manipulated”. I guess, they mean modulation or stabilization but I am not sure. Since mysterious manipulation seems to be essential for understanding the whole work, it should be defined precisely and unambiguously.
  • In the “Methods” the authors describe the preparation and usage of the HTHP, 300-micrometer NV sample but provide almost no information on the other CVD bulk sample, except for the irradiation and annealing details. They also do not provide any information on how the NV sample was attached to the fiber. In particular, nothing is mentioned about how far was the microwave coil from the sample. There is no information on how long is the fiber and whether it plays any relevant role in the measurement except for transmitting light to and from the NV sample. Figure 1c with its block scheme is not sufficient to see how the experiment is conducted.
  • In “Results”, the authors present results of calibration of the temperature dependences of two samples, which differ quite strongly. Some discussion of this difference would be appropriate here. Moreover, the fitted curve is visibly different from the measured values but again no comment or explanation is provided and the authors write that the “overall trend was essentially the same”. Speaking of the temperature dependences it would be interesting to comment if the offset of two calibration curves (for CVD and HTHP samples) is characteristic of the diamond production or is it just the usual scatter of sample parameters.
  • The most important part of the manuscript, Subsection 3.3 is not sufficiently detailed. Here, the mysterious word “manipulation” is causing the most serious comprehension problems, but also Figure 4 includes a number of undefined symbols and has neither adequate explanation nor sufficiently detailed caption. Also, it looks as if Eq. (2) describes the fluorescence change, which is defined as symbol S, whereas it seemed to me that earlier the fluorescence was defined as capital F (I’m writing, “it seemed” since it was not clearly defined).
  • Besides the above-mentioned problems, the manuscript is very difficult to understand because of the bad language, far from the correct scientific English.

In summary, despite the manuscript presents some interesting, albeit not very original, results I cannot recommend its publication.

Round 2

Reviewer 1 Report

The Authors need to explain clearly what this publication adds to the field. 
As I read it, the demonstration of temperature compensation for NV fibre magnetometers is the key result presented in this work.  However, in a paper cited by the authors (Hatano (2021), Applied Physics Letters, 118 034001) simultaneous thermometry and magnetometry with an NV fibre coupled magnetometer is already demonstrated.  Its not clear at all what this work adds to that demonstration.

Author Response

Thank you very much for the question. We would explain the difference more clearly between our work with Hatano (2021), Applied Physics Letters, 118 034001.

They introduced a centimeter-scale portable quantum sensor head, where the fiber was fixed to a surface waveguide microwave transmitter. The design of the sensor probe, including the surface waveguide microwave transmitter and its connection method with the fiber, weakened the size and portability advantage of the fiber scheme. In the work, continuous measuring mode is adopted to analyze the noise from the magnetic filed spectrum and test the long-term stability under a magnetic shielding condition, which was done in a laboratory environment.

We adopted diamond particles fixed on the fiber tip suppressing the sensor size to several millimeters. The more compact and portable microwave transmission methods we used are more likely to be used in a variety of practical situations. At the same time, the microwave transmission method can also be optimized in the subsequent work through the microfabrication and other ways to improve the consistency of the measurement system. Furthermore, we had directly analyzed the temperature noise caused by laser and microwave system in the magnetic measurement system, which is more helpful to solve the related practical application problems. Those temperature noises were compensated by the pulsed magnetic field measurement with multi-frequency synchronous manipulation, which could be used in practical sensing applications. We believe that this agrees with the requirements of optical fiber sensor research in Sensors.

We really hope to improve our understanding and contribution in this field through related work. Thank you sincerely for your patience and guidance!

Reviewer 2 Report

The re-submitted manuscript is significantly improved with respect to the previous version and,  in principle, I would be ready to recommend its publication.

However, I still object to calling the described type of NV magnetic sensor a “fiber magnetometer”. The authors replied that, like some other authors, they use the definition in a broad sense and did not change the sensor name.

I strongly disagree with their decision. It is not just a matter of using a more flexible or more general definition. It is the issue of being either precise and unambiguous or otherwise. Using the name “fiber magnetometer” for the device in which an optical fiber is used only for signal transmission and not as an intrinsic part of the sensor is misleading and incorrect. The fact that other authors (and their editors with reviewers!) approve that incorrect nomenclature simply indicates that the bad habit tends to spread, rather than warrants its correctness.

The authors could at least add a paragraph for the readers’ convenience explaining that various kinds of “fiber magnetometers” may exist and extend appropriately the reference list.

In conclusion, I cannot recommend publication of the resubmitted version as long as the above-mentioned issue is not fixed. 

Author Response

Thank you very much for the suggestion. After carefully consideration and more literature research, we agree that the bad habit of incorrect naming tends to spread. We have changed the title to ‘Temperature Fluctuations Compensation with Multi-Frequency Synchronous Manipulation for a NV Magnetometer in Fiber-optic Scheme’, hoping for a more accurate description.